# Artificial Intelligence in Ultrasound Diagnoses of Ovarian Cancer: A Systematic Review and Meta-Analysis

**DOI:** 10.3390/cancers16020422

**Published:** 2024-01-19

**Authors:** Sian Mitchell, Manolis Nikolopoulos, Alaa El-Zarka, Dhurgham Al-Karawi, Shakir Al-Zaidi, Avi Ghai, Jonathan E. Gaughran, Ahmad Sayasneh

**Affiliations:** 1Department of Women’s Health, Guy’s and St Thomas’ Hospital NHS Foundation Trust, London SE1 7EH, UK; 2Department of Gynaecology, Alexandria Faculty of Medicine, Alexandria 21433, Egypt; 3Medical Analytica Ltd., Flint CH6 SXA, UK; 4School of Life Course Sciences, Faculty of Life Sciences and Medicine, King’s College London, Strand, London WC2R 2LS, UK; 5Department of Gynaecological Oncology, Surgical Oncology Directorate, Cancer Centre, Guy’s Hospital, Great Maze Pond, London SE1 9RT, UK; 6School of Life Course Sciences, Faculty of Life Sciences and Medicine, St Thomas Hospital, Westminster Bridge Road, London SE1 7EH, UK

**Keywords:** machine learning, artificial intelligence, ultrasound, ovarian cancer, ovarian tumours

## Abstract

**Simple Summary:**

According to cancer research statistics, there are 7500 new ovarian cancer diagnoses in the UK each year. An earlier detection of ovarian cancer leads to a better prognosis; however, there is currently no screening programme for ovarian cancer, and detection using ultrasound examinations remains challenging. The use of artificial intelligence in gynaecological ultrasound examinations aims to improve the diagnostic accuracy of ultrasound for ovarian cancer and improve outcomes for patients. This review aims to collate current research on AI in the ultrasound diagnosis of ovarian cancer and suggests the usefulness of incorporating this into clinical care.

**Abstract:**

Ovarian cancer is the sixth most common malignancy, with a 35% survival rate across all stages at 10 years. Ultrasound is widely used for ovarian tumour diagnosis, and accurate pre-operative diagnosis is essential for appropriate patient management. Artificial intelligence is an emerging field within gynaecology and has been shown to aid in the ultrasound diagnosis of ovarian cancers. For this study, Embase and MEDLINE databases were searched, and all original clinical studies that used artificial intelligence in ultrasound examinations for the diagnosis of ovarian malignancies were screened. Studies using histopathological findings as the standard were included. The diagnostic performance of each study was analysed, and all the diagnostic performances were pooled and assessed. The initial search identified 3726 papers, of which 63 were suitable for abstract screening. Fourteen studies that used artificial intelligence in ultrasound diagnoses of ovarian malignancies and had histopathological findings as a standard were included in the final analysis, each of which had different sample sizes and used different methods; these studies examined a combined total of 15,358 ultrasound images. The overall sensitivity was 81% (95% CI, 0.80–0.82), and specificity was 92% (95% CI, 0.92–0.93), indicating that artificial intelligence demonstrates good performance in ultrasound diagnoses of ovarian cancer. Further prospective work is required to further validate AI for its use in clinical practice.

## 1. Introduction and Background

Ovarian tumours have a variety of differential diagnoses, from benign and borderline to malignant tumours, and the origin of these is primarily gynaecological [1]. Ovarian cancer has the worst prognosis among all gynaecological cancers and is often diagnosed in its most advanced stages due to its vague presenting symptoms and due to the lack of an effective screening programme for ovarian cancer [2]. In addition to this, there are large geographical variations in the incidence of ovarian cancer, with a higher incidence in developed countries [3]. The survival rates for malignant tumours vary depending on the stage of the disease, with the stage at diagnosis and the pathological subtype of ovarian malignancy being some of the most important factors for survival [2]. Stage 1 ovarian cancer has a 5-year survival rate of greater than 90% compared to stage 4 ovarian cancer, which carries a 5-year survival rate of 13.4% [2]. High-grade serous carcinoma account for a significant proportion of ovarian cancer diagnoses and are thought to have the worst prognoses, as they are clinically aggressive tumours [4].

The treatments offered to patients can differ significantly based on the provisional diagnosis from imaging modalities, prior to a confirmative histological diagnosis [5]. Benign masses can be monitored without intervention, or, depending on their size and symptoms, a limited resection with a low peri-operative risk can be performed and may be beneficial for patients desiring fertility preservation [6,7,8]. It is known that findings of unilocular cysts in premenopausal women that measure less than 5 cm are associated with a less than 1% risk of malignancy, and these can be managed conservatively [9]. Malignancy risk increases with a family history of ovarian and breast cancer, postmenopausal status, and solid components on scans [9]. Ovarian malignancy, by contrast, requires treatment in gynae-oncology centres with appropriate staging, and radical surgery can be offered [10]. A systematic review demonstrated that the outcomes for patients with ovarian cancer are improved when treatment is provided by gynae-oncologists in specialised centres [11]. The accurate pre-operative characterisation of ovarian tumours is therefore crucial to improving patients’ outcomes and reducing the morbidity burden of the disease [12].

Ultrasound, magnetic resonance imaging (MRI), and computer tomography (CT) have traditionally been used for the assessment of these tumours. CT is mainly used for the preoperative staging of disease and disease follow-up in ovarian cancer, and MRI can be used as an adjunct staging method when an ultrasound examination is uncertain or, additionally, to stage the disease preoperatively [13]. Ultrasound is the first-line imaging modality for the assessment of ovarian tumours, as it is easily accessible and cheaper than other alternatives, and, when interpreted by expert examiners, it has been shown to have excellent results for distinguishing between benign and malignant tumours [12]. The limitations of all imaging modalities, especially those of ultrasound examinations, include operator dependency, and this can have a significant impact on the outcomes of patients [14,15]. To reduce interobserver variability, multiple risk stratification models were introduced with excellent diagnostic accuracy. These include the risk of malignancy index (RMI), which is calculated using the serum Ca-125, the ultrasound scan features, and the menopausal status, and is useful as a discriminant between benign and malignant tumours [16]. The International Ovarian Tumour Analysis (IOTA) models for ultrasound include logistic regression models that use the Ca-125 and ultrasound features such as locularity, echogenicity, and colour Doppler and the Ovarian–Adnexal Reporting and Data System (O-RADS) MRI can categorise tumours in five categories and estimates the risk of malignancy from 0 to 90% [17,18,19].

However, with the aim of further improving the diagnostic accuracy of ultrasound examinations for ovarian tumour diagnoses and reducing interobserver and intraobserver bias, there has been increased interest in the use of computer-aided diagnosis (CAD) for ultrasound image analyses [20].

Artificial intelligence using radiomic techniques have the ability to extract large numbers of quantitative imaging features from ultrasound images that can determine the aetiology of an ovarian tumour. It is a multistep process that involves choosing a region of interest (ROI) from an ultrasound image, followed by the extraction of shape, textural, and statistical features from the images. These data are then input into machine learning tools, which can then give information on a desired outcome, such as tumour diagnosis [21]. Such methods have demonstrated excellent diagnosis performance for ovarian tumours. Acharya et al. reported a modality that used 11 significant features in k-nearest neighbour (KNN) and probabilistic neural networks (PNNs), two supervised learning-based classifiers with 100% classification accuracy, sensitivity, specificity, and positive predictive value in detecting ovarian cancer [22]. Further promising performance results have been published in other studies [23,24,25].

Here, we present a systematic review and meta-analysis of published data on the performance of artificial intelligence diagnostic algorithms for the ultrasound diagnosis of ovarian tumours.

## 2. Review

### 2.1. Methods

Embase and MEDLINE were searched from the beginning of each database until October 2022 for original studies that developed or used AI algorithms in diagnosing ovarian cancer through ultrasound images. This aimed to contribute meaningful insights to this field, and the extended timeframe allowed for the inclusion of a broad spectrum of studies, capturing the evolution of AI algorithms. Embase and MEDLINE were chosen as search databases, as these are two of the most prominent databases in the field of medical and biomedical literature. These two databases are often cited together and offer comprehensive coverage with the use of the MeSH framework, ensuring a high level of indexing.

Titles, abstracts, and MeSH terms were searched for combinations of the following words: artificial intelligence (texture analysis, texture feature, deep learning, machine learning, neural network); ultrasonography (ultrasound, ultrasound image, transvaginal, sonogram, transabdominal, imaging); ovarian cysts (ovarian mass, ovarian tumour, ovarian tumour, neoplasm, malignancy, ovary). Only articles written in English were included in the systematic analysis. This review adhered to the Preferred Reporting Items for Systematic Reviews and Meta-Analyses (PRISMA). Following the established guidelines improved transparency and ensured that a high-quality systematic review was maintained. This review was not registered. While registration is recommended, the absence of formal registration does not negate the value of a review and provides context for readers to consider.

### 2.2. Inclusion and Exclusion Criteria

The studies included were those that explicitly analysed the processing and performance of AI algorithms in the ultrasound diagnoses of ovarian tumours and had final histological results, which are considered the gold standard for diagnosis in human subjects. This ensured that there was a focus on the unique challenges associated with ultrasound diagnoses of ovarian tumours using AI algorithms and that the analysis was on imaging data, with the selected studies only directly assessing the accuracy of AI algorithms in real-world clinical scenarios. This was the primary objective of this review. The articles were restricted to those only written in English to ensure a consistent language foundation and allow for a comprehensive understanding and synthesis of the selected studies. Only the studies where a 2 × 2 table to calculate sensitivity and specificity could be constructed were included.

The exclusion of reviews, editorials, and articles discussing the diagnostic performance of AI on histopathological images was crucial for maintaining focus on ultrasound imaging and ensured the review was dedicated to the primary imaging modality of interest.

### 2.3. Initial Screening

Two authors (SM and MN) conducted the initial screening of the titles and abstracts for suitability. This collaborative screening process ensured a comprehensive evaluation of relevance studies early in the review process and ensured that only the studies that met the inclusion criteria were included. To allow for a more in-depth assessment of the selected studies, full manuscripts were reviewed by SM and MN if deemed appropriate during the initial screening. Study characteristics and diagnostic performances of the AI models were extracted independently to ensure the reliability and objectivity of information gathered from the selected studies. For each of the eligible studies, the author’s name, year of publication, number of test sets used, type of US image (2D or 3D images) and type of US performed to collect the images (transabdominal, transvaginal, or both), sizes and number of images used in the database, type of database (prospective or retrospective), and diagnostic accuracy data were collected. Any disagreement regarding the articles were discussed with a third author, AS, until an agreement was reached. This ensured a fair and objective approach, further improving the reliability.

The inclusion and exclusion criteria, review process, database search strategy, methodological approach, and overall objectives outlined a comprehensive and rigorous systematic review methodology. This approach ensured the selection of relevant studies and gives valuable insight into the application of AI algorithms in diagnosing ovarian cancer from ultrasound images.

### 2.4. Statistical Analysis

The number of ultrasound images examined and the diagnostic accuracy data (true positive, false positive, true negative, and false negative) were extracted from the studies and used to construct 2 × 2 tables to calculate sensitivity and specificity with 95% confidence intervals (95% CI). The CI was assumed to be 95% with a *p* value of <0.05 for each study to have significant results. The upper and lower limits of the CI were calculated automatically by software based on 95% CI. The estimated sensitivity and specificity from each study were included in a meta-analysis using the Cochrane RevMan 5.4 software (Review Manager (RevMan) (Computer Programme), Version 5.4, The Cochrane Collaboration, 2020 [26]. This meta-analysis used a single test accuracy assessment without an added covariance matrix.

A weight-based forest plot was constructed using RevMan 5 to visually depict each result and demonstrate the combined results with weights. Neither the odds ratio nor the risk ratio were shown as the forest plot depicted the summary of the AI in the ultrasound diagnoses of ovarian cancer. A summary receiver operating characteristic curve (sROC) was also plotted with the ROC curve used to estimate the overall accuracy of AI in USs in detecting ovarian cancer. The area under the curve (AUC) determined the overall result and was denoted as >80%. Each of these studies has been represented with a circle, and the circle size represents the weight of the study based on sample size. The location of the circle is an estimate of the individual study sensitivity and specificity. The summary sensitivity and specificity were denoted as the largest circle, circle 19. The Preferred Reporting Items for Systematic Reviews and Meta-Analysis (PRISMA) was used for the results and discussion elements of this review.

## 3. Results

### 3.1. Prisma Flow Chart

The electronic search of the two databases initially yielded a total of 3726 papers. Crosschecking the references did not identify any additional papers. After the exclusion of duplicates and unsuitable papers based on title screening, a total of 63 abstracts were suitable for abstract screening. Following abstract screening and assessing the suitability of the abstracts, 32 full papers were accessed and analysed. A further 18 papers were excluded from the final analysis for the following reasons: not written in English [2]; did not meet the eligibility criteria [14]; duplicate [1]; and results redacted from publication [1]. Therefore, a total of 14 papers were included in this review, including 2 prospective studies and 12 retrospective studies (see Figure 1 and Table 1). Of these studies, 8 reported the use of only transvaginal US, 1 reported the use of transabdominal US, and 5 reported the use of both TVUS and TAUS. The type of AI and type of learning that the models used is documented in Table 2.

### 3.2. Meta-Analysis Results

All studies were used to calculate the number of ultrasound images included in the meta-analysis and the accuracy of the pooled diagnostic performances. Overall, 15,358 B-Mode ultrasound images were examined in the included studies. Studies from which a 2 × 2 table could be constructed were included in the final meta-analysis, with sensitivities ranging from 40% to 99% and specificities ranging from 76% to 99%. The overall pooled sensitivity was 81% (95% CI, 0.80–0.82), and the overall specificity was 92% (95% CI, 0.92–0.93). A forest plot was constructed, demonstrating the sensitivities and specificities of all the included studies, shown in Figure 2. Figure 3 demonstrates the ROC curve for the diagnostic utility of AI models diagnosing ovarian tumours on ultrasound images.

## 4. Discussion

With the continuously evolving use of AI in the diagnosis of ovarian cancer, there have been numerous studies analysing the diagnostic performance of different AI models. The definitive diagnosis of ovarian tumours is histological diagnosis, and AI has the potential to help in the prediction of pre-operative diagnoses and thus avoid unnecessary intervention [36]. The use of AI could further support the triaging of patients to the appropriate centres, improving survival rates, and could help avoid the over-treatment of benign masses, thus reducing the morbidity associated with this [37]. This is the first systematic review to assess AI’s diagnostic performance for ovarian cancer using ultrasound as the only imaging modality and ultrasound image information as the only input data for these models, with an extensive search of the literature and study databases.

Xu et al. reported a systematic review and meta-analysis of image-based ovarian cancer identification, including 17 ultrasound articles with a pooled sensitivity and specificity of 91% and 87%, respectively [38]. The pooled rates included in this present review were lower. A reason for this may be that our review included a total of 14 papers, and one of the studies included in the previously published systematic review has since been redacted [39]. In addition, in studies where more than one test set was used for the final validation of the model, we chose to include these in our meta-analysis. This increased the total number of US images analysed but may have influenced the overall diagnostic accuracies recorded compared to Xu et al.

In some studies, the same US images were used, but different features were extracted from them. Acharya et al. reported the use of 1300 benign and 1300 malignant 3D images using 11 features in the k-nearest number and the probabilistic neural network (PNN) classifiers to construct their Gynaescan [22]. This study demonstrated a 100% accuracy classification accuracy in detecting ovarian tumours. In a prior study, using the same dataset, 23 features were extracted from the 3D images including Hu invariant moments, Gabor wavelet features, and entropies, with a sensitivity of 99.2% and specificity of 99.6%—different diagnostic performances to the other study [35]. Both these studies were included separately in our meta-analysis, and—as different features were extracted from the 3D images, producing different accuracies—this contributed to the strength of this meta-analysis. These two studies were the only studies using 3D images and had a higher diagnostic performance than other studies included, perhaps demonstrating an increasing role in 3D imaging in AI models to improve diagnostic performance. However, it is important to note that access to 3D imaging is not widespread, and thus the generalisability of these results may be limited.

Comparison between different studies is difficult due to the heterogeneity in methodology techniques from the features extracted from the images to the segmentation techniques used. For example, segmentation techniques differ, with Chiappa et al. reporting the use of the TRACE4 radiomic segmentation tool in the AROMA study [24] compared to manual segmentation used by Al-Karawi et al. [20,28]. In addition to this, the AROMA study classified the ovarian masses into three groups: solid, cystic, and motley and reported diagnostic accuracies separately compared to the other studies where ovarian masses were divided into benign and malignant.

There is uncertainty regarding the quality of the ultrasound images included in the individual studies and the meta-analysis. Research has consistently demonstrated that TVUS is superior to TAUS for imaging adnexal masses [40,41,42]. The studies included here used a mix of TV and TA ultrasound images, with six studies recording the use of transabdominal ultrasound but not mentioning quality assurance or the ultrasound machines that were used to capture the images. While this could allow for the generalisability of our findings due to the global variations in ultrasound technology, quality images may improve the diagnostic performance of models and thus improve the overall accuracy of the AI models.

This review focussed solely on the diagnostic performance of AI models in US images. In clinical practice, there are several US tools that can improve the diagnostic accuracy of ultrasound examinations for ovarian cancer, and these can include other demographic features and characteristics of the patient that are pertinent in making a diagnosis [18,43,44,45]. These features can often be elicited from history, taking, for example, patient symptoms and family history; given this, there may be a place for AI tools used in clinical practice to consider such factors in their algorithms.

To ascertain the clinical value of AI in ultrasound diagnoses of ovarian masses, a direct comparison between expert examiner interpretation by pattern recognition and the use of such models is required on the same dataset to further validate the clinical role of AI within ultrasound diagnoses of ovarian cancer. Gao et al. demonstrated that the deep convolutional neural network model had increased accuracy and sensitivity compared to radiologists alone (87.6% vs. 78.3%, *p* < 0.0001), and evaluations for radiologists were increased when assisted by DCNN [32]. Such information on the models gives it clinical context, and all models should aim to be validated against expert examiners.

Also, within this field of research, many studies are retrospective in nature (12/14 included in this review); for further validation and robust testing, there is a need for prospective studies. Prospective studies would assess AI use in live clinical settings and thus assess its validity for use in such clinical scenarios, although this would most certainly have to be blinded to not influence the clinician’s decision making while training these models.

To date, the results of studies analysing the use of AI in ultrasound diagnoses of ovarian cancer often demonstrate the difference in accuracy between the diagnosis of benign and malignant conditions, and very few studies separate borderline ovarian tumours (BOTs) from malignant tumours in their analyses. This is an important point to raise, given the difference in management between these two tumour groups and the potential implications for a patient’s fertility. BOTs continue to pose a challenge for pre-operative diagnosis, and the accuracy of their diagnosis has been reported to be lower than that of both benign and malignant tumours on ultrasound images [46,47].

## 5. Future Directions

As AI is an emerging field within gynaecology, there is a potential for inherent bias in research methodologies, attributable to publication bias due to funding from private companies. To prevent such occurrences, the development of a core outcome set for computer-based systems in ultrasound diagnoses and stringent guidelines on methodology are required. This will improve consistency between studies and allow for a more effective comparison between results.

There are also very few studies at present that can use AI to differentiate between specific ovarian histologies. Some ovarian masses are known to have specific discriminating ultrasound characteristics that define them—for example, the acoustic shadowing present behind the papillary projection in serous cystadenofibromas [1]. It is also prudent for all future studies in AI in ovarian cancer diagnosis to document and report the exact histological subtypes that constitute the database used. Some centres and countries may have a small incidence of certain subtypes; for example, Asian nations have a higher rate of clear-cell and endometrioid ovarian cancers and a lower proportion of serous carcinomas compared to the rest of the world [48].

As the use of AI technology in ultrasound diagnoses of ovarian tumours progresses, there is a potential to develop technology that can be in used in the preoperative staging of ovarian cancer and in the intraoperative identification of structures such as lymph nodes. Fischerova et al. have reported studies where ultrasound technology has been used with good diagnostic accuracy in the preoperative assessment of different peritoneal compartments, retroperitoneal and inguinal lymph nodes and to assess the depth of rectosigmoid wall infiltration [49]. At present, this type of staging is not accessible in all centres; however, as ultrasound skills progress, there is the potential that the use of such skills with the help of AI technology could improve the staging of tumours and reduce the requirement for further imaging. Intraoperatively, the quick identification of affected lymph nodes and peritoneal compartments with ultrasound technology could possibly reduce operative time and its associated complications.

In addition to this, it is important that any ultrasound database used to train AI in a medical setting is from an ethnically diverse population to ensure the results are generalisable. Due to the increased interest in ovarian masses in pregnancy [50], training models with images from this group of patients should be considered.

## 6. Limitations

There are a few limitations of this review to note. The review was not registered. We, as a research team, decided not to register the review, as our review had a narrow scope, focussing solely on the AI in the ultrasound characterisation of ovarian cancer; as mentioned, there have been previous reviews that focussed on AI in the imaging diagnosis of ovarian cancer. As our review gives an updated and more focussed approach than the previous reviews, it was not registered. However, we understand that registration is advised, and this could be considered a limitation. In addition to this, this is a small volume systematic review with 14 studies included. The small numbers of studies and heterogeneity between AI studies make direct comparison challenging, and this could be a limitation. To ensure an in-depth comprehension of the studies, only studies written in English were included in this review, and this could include a language bias. The development of AI tools used in US diagnoses of ovarian cancer could have been documented in other languages, but, because of our strict inclusion criteria, they would have been excluded.

## 7. Conclusions

In conclusion, artificial intelligence has great potential to improve the pre-operative diagnosis of ovarian cancer on ultrasound examinations, with promising results demonstrated thus far in the literature. The decision making is ultimately still the responsibility of the clinician, but a synergistic relationship between the clinician and AI could facilitate earlier diagnosis and continue to improve outcomes.

## Figures and Tables

**Figure 1 cancers-16-00422-f001:**
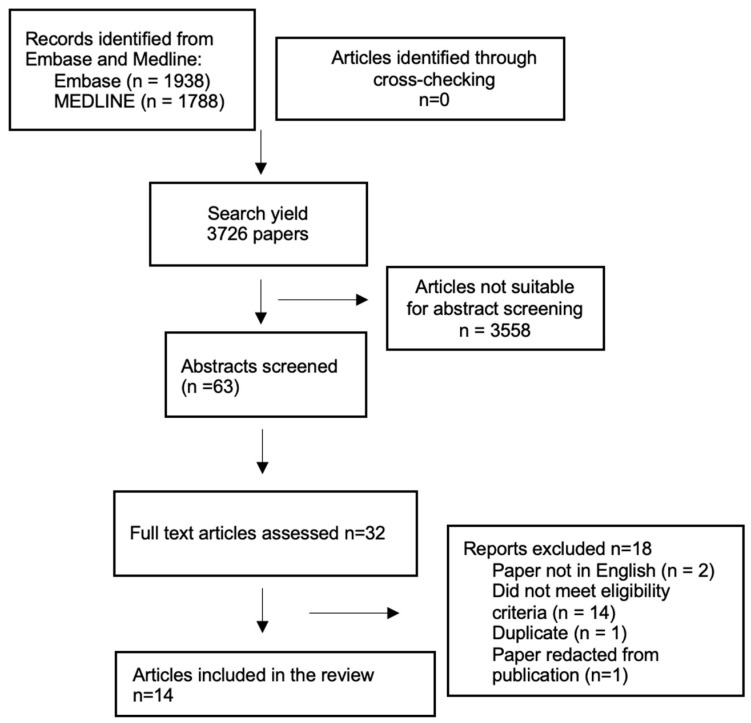
Preferred Reporting Items for Systematic Reviews and Meta-Analysis (PRISMA) flow chart.

**Figure 2 cancers-16-00422-f002:**
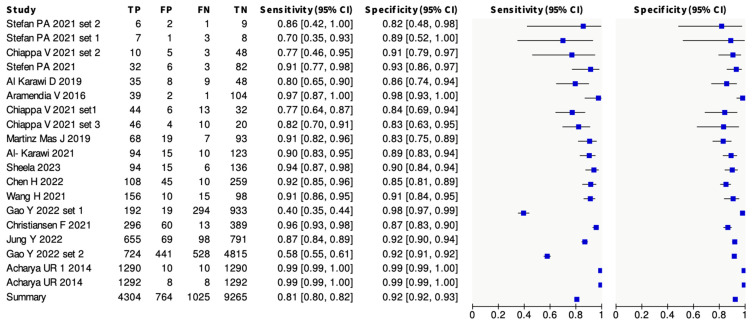
Forest plot for the 14 included studies: Stefan et al., 2021 [27]; Chiappa et al., 2021 [24]; Al-Karawi et al., 2019 [28]; Aramendia V et al., 2016 [23]; Martinez Mas et al., 2019 [29]; Al-Karawi et al., 2021 [20]; Sheela et al., 2022 [30]; Chen H et al., 2022 [31]; Wang et al., 2021 [25]; Gao Y et al., 2022 [32]; Christiansen F et al., 2021 [33]; Jung Y et al., 2022 [34]; Acharya et al., 2014 [22]; Acharya et al., 2014 [35]. Heading abbreviations are as follows: TP—True positive, FP—False positive, FN—False Negative, and TN—True Negative. Overall sensitivity and specificity are demonstrated here, and neither odds ratio nor risk ratio were used.

**Figure 3 cancers-16-00422-f003:**
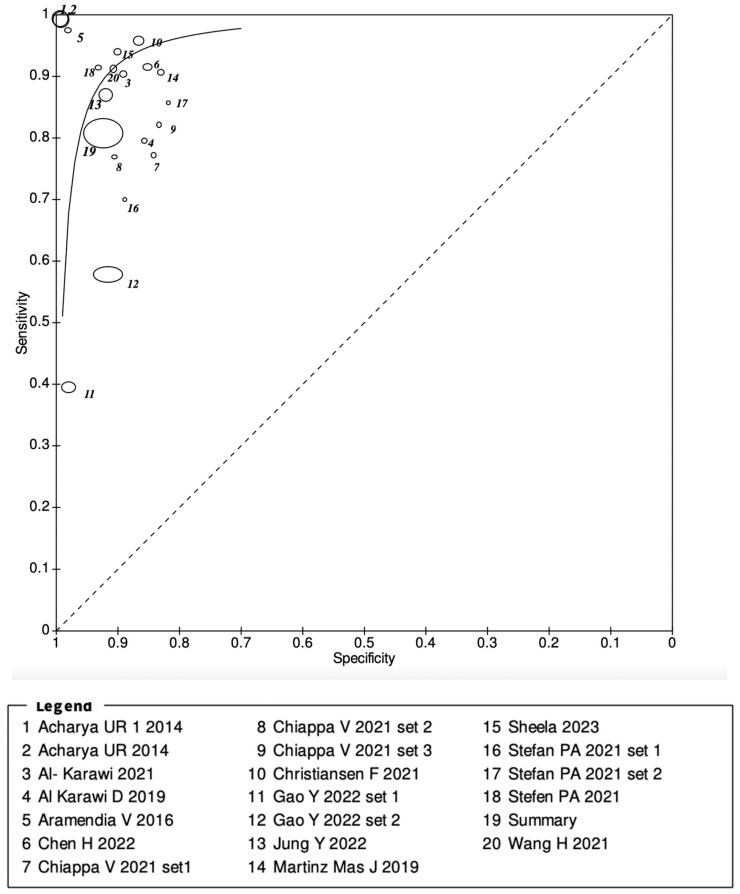
Summary receiver operating characteristic (SROC) curve for the 14 included studies: Stefan et al., 2021 [27]; Chiappa et al., 2021 [24]; Al-Karawi et al., 2019 [28]; Aramendia V et al., 2016 [23]; Martinez Mas et al., 2019 [29]; Al-Karawi et al., 2021 [20]; Sheela et al., 2023 [30]; Chen H et al., 2022 [31]; Wang et al., 2021 [25]; Gao Y et al., 2022 [32]; Christiansen F et al., 2021 [33]; Jung Y et al., 2022 [34]; Acharya et al., 2014 [22]; Acharya et al., 2014 [35]. Area under the curve (AUC) is demonstrated as 0.87.

**Table 1 cancers-16-00422-t001:** Studies included in the meta-analysis; design and type of ultrasound used.

Study	Design	Type of Ultrasound
Stefan et al., 2021 [27]	Retrospective	Transvaginal
Chiappa et al., 2021 [24]	Retrospective	Transvaginal
Al-Karawi et al., 2019 [28]	Prospective	Transabdominal and transvaginal
Aramendia V et al., 2016 [23]	Prospective	Transvaginal
Martinez Mas et al., 2019 [29]	Retrospective	Transabdominal and transvaginal
Al-Karawi et al., 2021 [20]	Retrospective	Transabdominal and transvaginal
Sheela et al., 2022 [30]	Retrospective	Transvaginal
Chen H et al., 2022 [31]	Retrospective	Transabdominal and transvaginal
Wang et al., 2021 [25]	Retrospective	Transabdominal
Gao Y et al., 2022 [32]	Retrospective	Transvaginal
Christiansen F et al., 2021 [33]	Retrospective	Transvaginal
Jung Y et al., 2022 [34]	Retrospective	Transabdominal and transvaginal
Acharya et al., 2014 [22]	Retrospective	Transvaginal
Acharya et al., 2014 [35]	Retrospective	Transvaginal

**Table 2 cancers-16-00422-t002:** Studies included in the meta-analysis; AI model used and type of learning.

Study	AI Model	Type of Learning (Machine or Deep)
Stefan et al., 2021 [27]	K-nearest number classifier (KNN)	Machine learning
Chiappa et al., 2021 [24]	Support vector machines (SVM)	Machine learning
Al-Karawi et al., 2019 [28]	Support vector machine (SVM)	Machine learning
Aramendia V et al., 2016 [23]	Multilayer perceptron network (MLP)/Neural network	Deep learning
Martinez Mas et al., 2019 [29]	K-nearest neighbours (KNN)/Linear discriminant (LD)/Support vector machine (SVM)/Extreme learning machine (ELM)	Machine learning
Al-Karawi et al., 2021 [20]	Support vector machine (SVM)	Machine learning
Sheela et al., 2022 [30]	Support vector machine (SVM)	Machine learning
Chen H et al., 2022 [31]	Residual network with two fusion strategies (feature and decision fusion)	Deep learning
Wang et al., 2021 [25]	Deep convolutional neural network (DCNN)	Deep learning
Gao Y et al., 2022 [32]	Deep convolutional neural network (DCNN)	Deep learning
Christiansen F et al., 2021 [33]	Deep neural network (DNN)	Deep learning
Jung Y et al., 2022 [34]	Deep convolutional neural network	Deep learning
Acharya et al., 2014 [22]	Probabilistic neural network (PNN), support vector machine (SVM), decision tree (DT), K-nearest neighbours (KNN), Naïve Bayes (NB)	Machine learning
Acharya et al., 2014 [35]	Probabilistic neural network (PNN)	Machine learning

## Data Availability

The data presented in this study are openly available in the MEDLINE and Embase databases.

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
