# Peer review of "Artificial Intelligence in Ultrasound Diagnoses of Ovarian Cancer: A Systematic Review and Meta-Analysis"

_cancers, 2024, doi:10.3390/cancers16020422_

Round 1
Reviewer 1 Report
Comments and Suggestions for Authors
The authors searched the database to do the systematic review and meta-analysis of literatures of artificial intelligence in ultrasound diagnosis of ovarian cancer. I had some suggestions in the following points.
1. In the "Results" section, I suggested the authors to describe the sonographic characteristics/features, the models or the algorithms for AI-aided diagnosis of ovarian cancer used in the selected literatures in detail. Summary of these items in a Table is good for the readers.
2. In Figure 2, what is TP, FP, FN and TN?
3. In line 178, "Xu et al. reported a systematic review and meta analysis on image based ovarian cancer identification, including 17 ultrasound articles with a pooled sensitivity and specificity of 91% and 87%, respectively." What is the reference?
4. In line 226, " Gao et al demonstrated that the Deep Convolutional neural network model had increased accuracy and sensitivity compared to radiologists alone (87.6% vs 78.3%, p <0.0001) and evaluations for radiologists were increased when assisted by DCNN." What is the reference?
Author Response
Thank you for consideration of our manuscript.
1. In the "Results" section, I suggested the authors to describe the sonographic characteristics/features, the models or the algorithms for AI-aided diagnosis of ovarian cancer used in the selected literatures in detail. Summary of these items in a Table is good for the readers – Thank you. – the models and algorithms have now been included in table labelled table 2. The precise ultrasound features extracted within each paper have not been included on the table as some studies have >200 ultrasound features included for each Region of Interest.
2. In Figure 2, what is TP, FP, FN and TN? – Thank you. True positive, false positive, false negative, true negative. This has now been added to the figure.
3. In line 178, "Xu et al. reported a systematic review and meta-analysis on image based ovarian cancer identification, including 17 ultrasound articles with a pooled sensitivity and specificity of 91% and 87%, respectively." What is the reference? – Thank you. This has been completed.
4. In line 226, " Gao et al demonstrated that the Deep Convolutional neural network model had increased accuracy and sensitivity compared to radiologists alone (87.6% vs 78.3%, p <0.0001) and evaluations for radiologists were increased when assisted by DCNN." What is the reference? – Thank you. This has been completed.
Reviewer 2 Report
Comments and Suggestions for Authors
This article provides a systematic review and meta-analysis of the application of artificial intelligence(AI) in the ultrasound diagnosis of ovarian cancer(OC).
Major concerns
1.Systematic reviews and meta-analyses should extensively include various databases. However, the author only searched two databases, Embase and MEDLINE databases,without mentioning the reasons,which the author should address.
2.The abstract of this manuscript lacks necessary explanations and has poor logical coherence. For example, in line 31, "Studies using histopathological findings as the standard were included," needs to specify the total number of studies. In line 37, "The overall sensitivity was 81% (95% Confidence Interval[CI], 0.80-0.82) and specificity was 92% (95% CI, 0.92-0.93), indicating that artificial intelligence plays an important role in aiding in the ultrasound diagnosis of ovarian cancer," only indicates AI models with good performance other than "plays an important role in...".
3.The usage of abbreviations requires refinement.
4.The introduction provides insufficient information about the staging of OC and the progress of AI in the diagnosis, especially early diagnosis (according to FIGO staging). The author needs to list more impressive developments in this field to highlight the role of AI in improving the management of OC patients.
5.The author mentions that this review is not registered. A clear reason must be provided for the absence of registration.
6.The basis for choosing a 95% CI value for statistical analysis needs clarification. The method and detailed information for calculating the 95% CI values should be explained.
7.The results section should follow with subsections and individual subtitles for each subsection.
8.Figure 3 lacks more detailed information. The meanings of the circle sizes in the figure are unclear and some of the numbers were overlapping. There were no specific steps for creating the figures.
Author Response
1.Systematic reviews and meta-analyses should extensively include various databases. However, the author only searched two databases, Embase and MEDLINE databases, without mentioning the reasons, which the author should address- Thank you. These are two of the most prominent databases in the field of medical and biomedical literature and are often cited together. A comprehensive coverage is possible, and the use of the MeSH framework ensures high quality o f indexing. This has now been included in the manuscript.
2.The abstract of this manuscript lacks necessary explanations and has poor logical coherence. For example, in line 31, "Studies using histopathological findings as the standard were included," needs to specify the total number of studies. In line 37, "The overall sensitivity was 81% (95% Confidence Interval[CI], 0.80-0.82) and specificity was 92% (95% CI, 0.92-0.93), indicating that artificial intelligence plays an important role in aiding in the ultrasound diagnosis of ovarian cancer," only indicates AI models with good performance other than "plays an important role in...". – Thank you. This has been amended.
3.The usage of abbreviations requires refinement – Thank you – the abbreviations in the manuscript have been checked.
4.The introduction provides insufficient information about the staging of OC and the progress of AI in the diagnosis, especially early diagnosis (according to FIGO staging). The author needs to list more impressive developments in this field to highlight the role of AI in improving the management of OC patients – Thank you. The introduction has been expanded extensively to include background on ovarian ca, staging, current diagnostic approach and the role of AI in improving OC diagnosis.
5.The author mentions that this review is not registered. A clear reason must be provided for the absence of registration.- Thank you. This is now discussed in the limitations section of the review.
6.The basis for choosing a 95% CI value for statistical analysis needs clarification. The method and detailed information for calculating the 95% CI values should be explained – Thank you. CI was assumed at 95% with a p value <0.05 for the study to have significant results. This is now included in the stats section.
7.The results section should follow with subsections and individual subtitles for each subsection thank you. This has now been completed.
8.Figure 3 lacks more detailed information. The meanings of the circle sizes in the figure are unclear and some of the numbers were overlapping. There were no specific steps for creating the figures. – Thank you. Each circle represents a study and the weight of the study is depicted by the size of the circle. The location of the circle is the estimate of the sensitivity and specificity with the summary sensitivity and specificity denoted by the largest circle. This has now been addressed in the stats section.
Reviewer 3 Report
Comments and Suggestions for Authors
Reviewer comments and suggestions
The authors in this study highlighted Artificial intelligence which is an emerging field within gynaecology and has been shown to aid in the ultrasound diagnosis of ovarian cancers.
The authors collected the studies from Embase and MEDLINE databases were searched and all original clinical studies that used artificial intelligence in ultrasound for the diagnosis of ovarian malignancies were screened. They have included fourteen studies were included in the final analysis, each of which had different sample sizes and used different methods. The result included that the sensitivity was 81% (95% CI, 0.80–0.82) and specificity was 92% (95% CI, 0.92–0.93), which may indicates artificial intelligence plays an important role in aiding in the ultrasound diagnosis of ovarian cancer
Overall, the manuscript is well written. I have listed the concerns and comments that needed to be explained or modified.
- Line 44-45 Please elaborate on this part for the common reader of your manuscript.
- Line 52-53 It's common information so I do not think it requires reference, if you specifically want to add up references then it need to be explained comprehensively.
- Line57 RMI first time used so it should be in full form
- Line 58-59 Please explain a bit about these models.
- Line 128-129 by which method (index) the authors used ROC curve
- Query related to figure 2 Did this plot show odd ratio or risk ratio?
- Comments for the first paragraph of the discussion, “It would be nice if the authors could add the novelty in the first paragraph of discussion”
- Line 188-189 What would be an inference the authors can provide by this study?
- What AI tools the authors used in the study needed to be discussed in the material and method part.
- Line 221-222 What the authors want to discuss here.
- It would be important to discuss the limitations of this study.
- Modify the References 33 and 41 based on the journal guidelines.
Author Response
- Line 44-45 Please elaborate on this part for the common reader of your manuscript. – Thank you. This has been completed.
- Line 52-53 It's common information so I do not think it requires reference, if you specifically want to add up references then it need to be explained comprehensively – Thank you. This sentence has been extended.
- Line57 RMI first time used so it should be in full form – Thank you. This has been amended.
- Line 58-59 Please explain a bit about these models – Thank you. This has been amended.
- Line 128-129 by which method (index) the authors used ROC curve – Thank you. This is has now been included in 180-181.
- Query related to figure 2 Did this plot show odd ratio or risk ratio? – Thank you. No, OR or RR was not included, and this is now written in the statistics section.
- Comments for the first paragraph of the discussion, “It would be nice if the authors could add the novelty in the first paragraph of discussion” – Thank you. The novelty of the manuscript is that this is the first review that uses ultrasound as the only imaging modality and input data into the AI models. This has now been included in the first paragraph.
- Line 188-189 What would be an inference the authors can provide by this study? – thank you. Apologies the relevance of this study was explained in the following paragraph with regards to extracting different features and the use of 3D imaging. This has now been put into one paragraph with the paragraph below.
- What AI tools the authors used in the study needed to be discussed in the material and method part. – Thank you. This has now been included in table 2.
- Line 221-222 What the authors want to discuss here – Thank you. To highlight that other features are important in making a diagnosis and AI tools may be require to include such history in their algorithms for clinical practice. This has now been included in the paragraph.
- It would be important to discuss the limitations of this study. Thank you, these have been included.
- Modify the References 33 and 41 based on the journal guidelines – Thank you. This has now been amended.
Round 2
Reviewer 1 Report
Comments and Suggestions for Authors
The authors addressed the comments. The article is acceptable for publication.
Reviewer 2 Report
Comments and Suggestions for Authors
I appreciate the author's efforts to revise the
manuscript.No further revision is needed.